# Subcutaneous Emphysema Related to Dental Treatment: A Case Series

**DOI:** 10.3390/healthcare10020290

**Published:** 2022-02-01

**Authors:** Rieko Shimizu, Shintaro Sukegawa, Yuka Sukegawa, Kazuaki Hasegawa, Sawako Ono, Ai Fujimura, Izumi Yamamoto, Soichiro Ibaragi, Akira Sasaki, Yoshihiko Furuki

**Affiliations:** 1Department of Oral and Maxillofacial Surgery, Kagawa Prefectural Central Hospital, 1-2-1, Asahi-Machi, Takamatsu 760-8557, Kagawa, Japan; de421021@s.okayama-u.ac.jp (R.S.); yuka611225@gmail.com (Y.S.); de421040@s.okayama-u.ac.jp (K.H.); sugar.x.48@gmail.com (A.F.); iyamamoto8408@gmail.com (I.Y.); furukiy@ma.pikara.ne.jp (Y.F.); 2Department of Pathology, Kagawa Prefectural Central Hospital, Takamatsu 760-8557, Kagawa, Japan; de19008@s.okayama-u.ac.jp; 3Department of Oral and Maxillofacial Surgery, Okayama University Graduate School of Medicine, Dentistry, and Pharmaceutical Sciences, Okayama 700-8525, Okayama, Japan; sibaragi@md.okayama-u.ac.jp (S.I.); aksasaki@md.okayama-u.ac.jp (A.S.); 4Advanced Research Center for Oral and Craniofacial Sciences, Okayama University Graduate School of Medicine, Dentistry, and Pharmaceutical Sciences, Okayama 700-8525, Okayama, Japan

**Keywords:** subcutaneous emphysema, dental treatment, computed tomography, 3D images

## Abstract

Cervicofacial subcutaneous emphysema (SE) is primarily caused by dental treatment introducing gas into the subcutaneous tissue. Air rapidly dissects into the subcutaneous tissue with face and neck swelling, leading to respiratory distress, patient discomfort, and chest pain. Computed tomography (CT) can detect spreading SE patterns. However, the true volume of SE and the degree of air changes in the body over time remain unknown. We evaluated the healing process of SE and the temporal changes in the volume of emphysema in three cases detected using our hospital’s electronic health record systems based on inclusion and exclusion criteria over the past 10 years, with CT and three-dimensional (3D) images. The first case was a 46-year-old woman who presented with complaints of swelling from her right eyelid to the neck and clavicles, pain on swallowing, respiratory distress, and hoarseness. The second case was a 35-year-old man who presented with complaints of swelling over the face. The third case was a 36-year-old man who presented with complaints of swelling from the left cheek to the neck. CT revealed SE and pneumomediastinum in all cases. All the patients were administered an antibacterial drug. The CT and 3D images showed an improvement in emphysema 3 days after the onset, with more than half of the volume reduction in emphysema. This made it possible to evaluate the changes in the air content of SE. Observation with CT until the healing process of SE is completed is crucial, and 3D images also help evaluate changes over time.

## 1. Introduction

Cervicofacial subcutaneous emphysema (SE) is known to be caused by the invasion of gas into the subcutaneous tissue. It is also caused by trauma such as facial fractures, intraoral trauma, traumatic destruction of the chest wall or the airway gastrointestinal tract, and dental treatment [1,2,3,4]. Jones A. conducted a literature review from 1993 to 2020 and found that dental extraction often preceded the onset of SE (54% of cases) in dental treatment. Most cases were iatrogenic, 51% were due to air turbines, 9% were due to air syringes, and factors such as nose-blowing accounted for 10%. When gas enters the subcutaneous tissue gap, its spread rate is high and is accompanied by the swelling of the face and neck, along with dysphagia, chest tightness, and dyspnea in some cases [5]. Computed tomography (CT) involves a small amount of exposure for the imaging, but it provides useful information for lesion location and differential diagnosis [6,7]. Using this CT feature to determine the extent of emphysema progression is useful for diagnosis [8]. However, no studies have been conducted on the true volume of SE and the degree of change in the air in the body over time and the correlation between the actual amount of SE and the progression of the disease. Understanding the course of healing of SE from the evaluation of volume over time is an important point that can serve as a guideline for future treatment. The first objective of this study was to observe the healing process of cervicofacial subcutaneous and mediastinal emphysema following dental treatment on a computer simulation using CT until resolution; the second was to evaluate the temporal changes in emphysema volume using 3D images based on computer simulation.

## 2. Case Series and Methods

This retrospective study analyzed the medical records of patients treated at the Department of Oral and Maxillofacial Surgery of the Kagawa Prefectural Central Hospital (Takamatsu, Japan) between April 2011 and July 2021. The design and methodology of this study were approved by the Ethics Committee of the Kagawa Prefectural Central Hospital (Approval No. 1072). The requirement for informed consent was waived by the ethics committee. This study conformed to the tenets of the Declaration of Helsinki.

Data regarding the diagnoses were obtained from our hospital’s electronic health record systems. Oral and maxillofacial surgeons with 7 and 15 years of experience perform CT interpretation of the head and neck area, and the chest and mediastinum were determined by a radiology specialist.

Thirty-six patients with SE were treated during the study period (Figure 1).

Inclusion criteria:SE caused by dental treatment;SE spanning multiple areas;CT was performed under appropriate conditions on the 1st and 3rd days of onset;The CT requirements for this research protocol were as follows:Use of a non-enhanced multidetector CT scanner (Aquilion; Toshiba Medical Systems, Tokyo, Japan);The CT acquisition parameters at our department of matrix were as follows: tube voltage 120 kVp; tube current 100 mA; 512 × 512 pixels; 1 mm slice thickness; 1 mm speed per rotation; 1 mm reconstructed slice increment; and 0° reconstruction algorithm bone gantry tilt;Performing a multi-section reconstruction on an image processing workstation using each parameter;Measurements being made on a high-resolution monitor of bone mode images;Exclusion criteria:SE of unknown causes;SE due to trauma, operation, thotacolaparotomy, and thoracentesis;CT at the second visit was performed the next day or more than 5 days after the initial visit;Visiting our hospital a few days after the injury.

In total, three eligible patients were enrolled in the study.

### 2.1. Range of Spread to Each Gap of SE

We classified either the face, neck, or mediastinum as the major categories to identify the extent of emphysema in each space.

The face comprises orbital soft tissue, buccal muscle, masticatory muscle, and the parotid gland. The neck includes the parapharyngeal, posterior pharyngeal, sublingual, and submandibular spaces. The mediastinum comprises anterior cervical, posterior cervical, posterior pharyngeal, and mediastinum, composed of the carotid artery space and anterior mediastinum.

The extent of emphysema was determined using CT on the 1st and 3rd days of illness.

### 2.2. SE Volume Measurement Method

Bone and aerial images were superimposed by 3D image construction with images obtained via CT as materials using maxillofacial surgery simulation software (Pro plan CMF 3.0, Materialize, Belgium).

The bone level and air thresholds were set to the ranges 226–3071 and −1024–−170, respectively.

The 3D image was converted into the stereolithography format to analyze the emphysema volume, and the volume was measured using xyz coordinates. A method for measuring the SE volume using 3D simulation from CT data is shown in the Appendix A.

The rate of reduction was calculated using the following formula:

(emphysema reduction rate) = (volume of emphysema measured from CT at follow-up)/(Volume of emphysema measured from the first CT) × 100

### 2.3. Case Series

#### 2.3.1. Case 1

A 46-year-old woman was referred to our hospital due to a sudden swelling of the face after endodontic treatment of the upper right lateral incisor. Root canal treatment was also combined with alternate irrigation using hydrogen peroxide. She presented with swelling, twisting sounds from her right eyelid and cheek to both neck and clavicles, pain on swallowing, respiratory distress in the prone position, and hoarseness (Figure 2). There were no obvious abnormalities in the oral cavity except the right maxillary lateral incisor, which was in the process of healing (Figure 3). The results of blood and biochemical tests revealed mild inflammation. CT performed the day after emphysema onset yielded a wide range of low-density images of foamy structures from the face to the mediastinum (Table 1 and Figure 4). An emergency hospitalization was proposed for strict follow-up, but the patient refused hospitalization. She was prescribed tablets containing amoxicillin hydrate 250 mg t.i.d for 3 days and was followed up daily at an outpatient clinic. On the 3rd day, CT images indicated a reduction in air volume (Figure 5), and the volume of air on the 3D image tended to decrease by more than half compared to the initial visit (Figure 6). Eight days later, the swelling of the face and neck was alleviated, the patient had fully recovered, and she returned to normal activities.

#### 2.3.2. Case 2

A 35-year-old man was referred to our hospital by a general dentist due to facial swelling observed during the extraction of the left mandibular wisdom tooth using an air turbine on the same day. He had swelling around the left orbit from the cheek to the neck and above the clavicle. Crepitus and snowball crepitation were observed at the swelling site upon palpation; however, no respiratory distress or chest symptoms were observed. Blood test results revealed a mild inflammatory reaction (CRP: 1.57 mg/dL), and CT showed a wide range of low-concentration images of foamy structures on the first day of admission (Table 1 and Figure 7C,D). Because the patient refused to be hospitalized, he was administered prophylactic levofloxacin hydrate (500 mg) orally o.d for 1 week and was followed up daily at an outpatient clinic. CT images taken 3 days later showed a decrease in the air (Figure 7E,F), and the air volume in the 3D image decreased by more than half compared to that observed during the initial visit (Figure 7A,B). Six days later, the swelling of the face and neck was alleviated, and the crepitus and snowball crepitation were no longer palpable.

#### 2.3.3. Case 3

A 36-year-old man was referred to our hospital by a general dentist because he presented with buccal swelling when his tooth surface was being cleaned with airflow after he underwent caries treatment of the right maxillary central incisor on the same day. Periodontal disease was also present in his oral environment. He had swelling from the left cheek to the neck. Blood test results showed no inflammation, and CT yielded several low-concentration images of foamy structures on the first visit (Table 1 and Figure 7C,D). He was hospitalized on the same day and started intravenous infusion of cefazolin sodium 1 g b.i.d for 3 days. CT images taken 3 days later revealed a reduction in the air volume (Figure 8E,F), and the air volume in the 3D image decreased by more than half compared to that observed during the initial visit (Figure 8A,B). His swelling was also alleviated.

## 3. Discussion

SE in the dental field is mainly caused by insufflation of air turbines, air syringes, carbon dioxide lasers., and firing with hydrogen peroxide solution during root canal treatment. Complications due to elevated pressure and facial fractures have also been reported [9,10]. SE can be easily palpated with crepitus and snowball crepitation as well as rapid swelling; however, there are complications that require a tracheostomy when mediastinal emphysema occurs due to the development of dyspnea [11].

Therefore, it is important to understand the extent of SE, including the presence or absence of mediastinal emphysema following dental treatment [12]. CT was performed either immediately after the onset or on the first and third days, but in cases 1 and 3, emphysema decreased mainly in the anterior cervical space by the third day after the onset. Volume analysis of the same compartment using a 3D model indicated that the reduction rate was approximately 75 and 100% in cases 1 and 3, respectively, and volume analysis of the face using a 3D model further confirmed that the reduction rate was approximately 63 and 77% in cases 1 and 3, respectively. Similar results were observed in the present study. This may be because the subcutaneous tissue in the neck is thought to absorb air faster than that in the face. Volume analysis using a 3D model also confirmed that the reduction rate was approximately 65, 52, and 79% in cases 1, 2, and 3, respectively, which was more than half. From this, it was considered necessary to take CT images up to the third day after the onset to confirm the time course of emphysema, exacerbation, and the presence or absence of relapse. The present study is the first report to our knowledge that uses CT to show the degree of iatrogenic air absorption due to emphysema in the body over time.

For a diagnosis of SE, it is necessary to make an appropriate differential diagnosis for infection, angioedema, and hematoma [13]. Careful recording of the dental history and confirmation of the major clinical characteristics of SE that differentiate it from other diseases are important for diagnosing SE related to dental treatment. Air that has invaded soft tissues often shows a cotton-like radiolucency even on plain radiographs, but CT is more useful for grasping the progress of that range, especially when it is suspected that it has spread to distant organs when it is better to take CT images actively.

As a treatment method for emphysema, it is sufficient to administer an antibacterial drug for infection prevention and rest, along with conservative therapy that involves waiting for the spontaneous absorption of emphysema, which is common [14]. Resting the patient through hospitalization management is desirable. If hospitalization is difficult, a lack of careful observation of the airways and non-monitoring of gas progression may lead to widespread emphysema and dyspnea due to airway compressions. In addition, if it spreads to the mediastinum, it may affect respiratory function and cardiac function, so caution is required [15]. In the case of SE extending to the mediastinum, symptoms such as chest pain and dyspnea will be observed due to tracheal and bronchial injury in the neck and chest, respectively, and the causative bacteria of dental infections invading the soft tissues together with emphysema. Regarding rest, antibiotics, and infection prevention, SE associated with dental procedures can cause the oral flora to invade the subcutaneous tissue due to the destruction of the oral mucosa, causing cellulitis or abscess formation during infection. Therefore, as with the administration of prophylactic antibiotics by dental treatment, the prophylactic administration of antibiotics is considered an essential first choice as a preventive measure against SE infection [2,16,17]. In this case, antibacterial agents were administered prophylactically until the symptoms improved, and emphysema disappearance was confirmed by CT. Good results were obtained without secondary infection. Furthermore, to prevent the inflow of additional air, it is necessary to avoid behaviors that increase oral pressure, such as forced exhalation, coughing, smoking, and gargling, and if abnormal symptoms appear after hospital discharge, follow-up at home is required. It is also important to note the need for a follow-up visit to the clinic.

This study had two limitations. The first is the number and timing of CT scans. In the present case observation study, CT was performed on days 1 and 3 after the onset of emphysema. Emphysema was still present on day 3, and its subsequent course to resorption is unknown. However, from a clinical standpoint, frequent CT scans in the absence of symptom exacerbation would be unnecessary from an ethical perspective. Second, few cases during dental treatment could be included as emphysema due to complications, and statistical evaluation was not possible. Therefore, future studies with several cases, such as multicenter studies, are required.

## 4. Conclusions

CT observation is vital until the healing process of cervicofacial subcutaneous and mediastinal emphysema, secondary to dental treatment, is completed. Evaluating the temporal change in emphysema volume using 3D images based on computer simulation, in addition to CT, helped to assess changes over time before SE disappeared.

## Figures and Tables

**Figure 1 healthcare-10-00290-f001:**
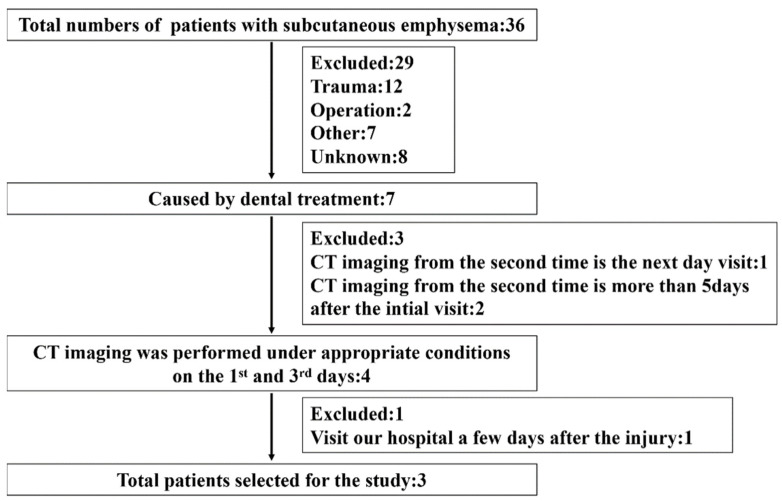
Flow diagram of the study representing the inclusion/exclusion criteria.

**Figure 2 healthcare-10-00290-f002:**
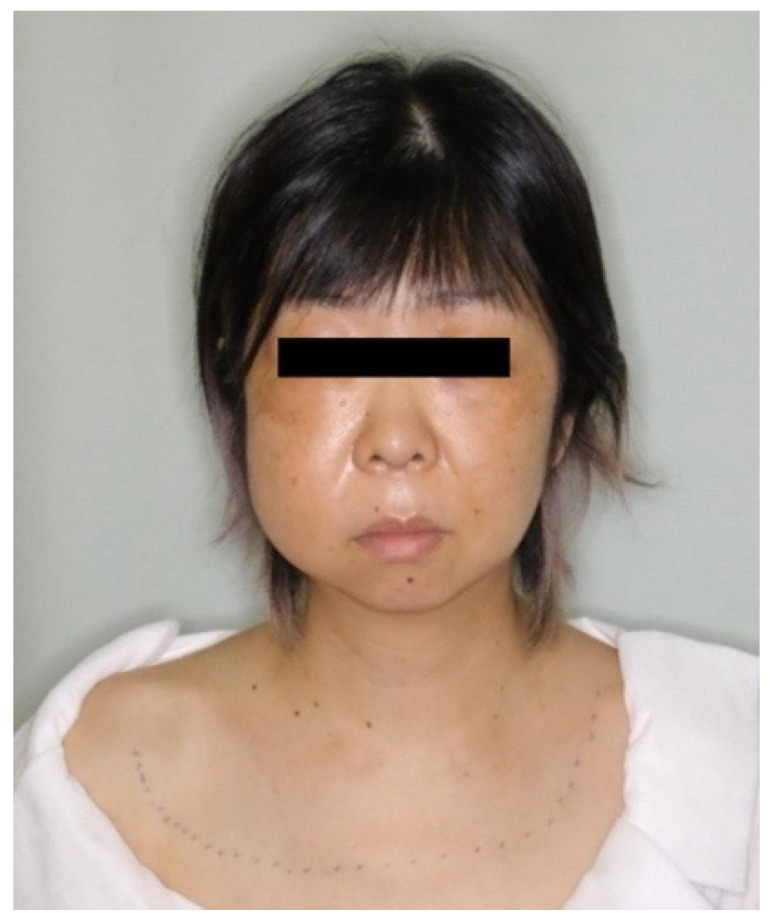
Patient (Case 1) had swelling from her right eyelid/cheek to both necks/clavicle area.

**Figure 3 healthcare-10-00290-f003:**
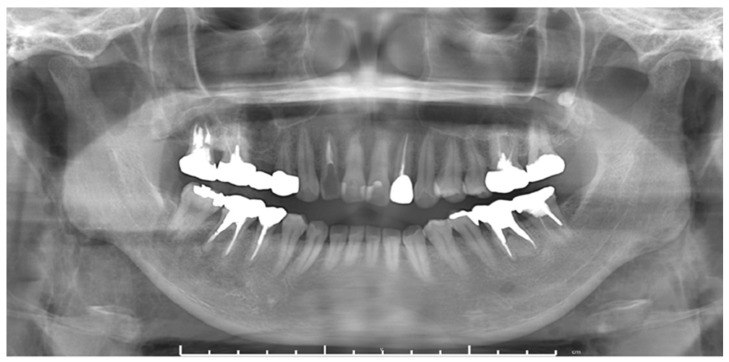
Panoramic radiograph at the first visit.

**Figure 4 healthcare-10-00290-f004:**
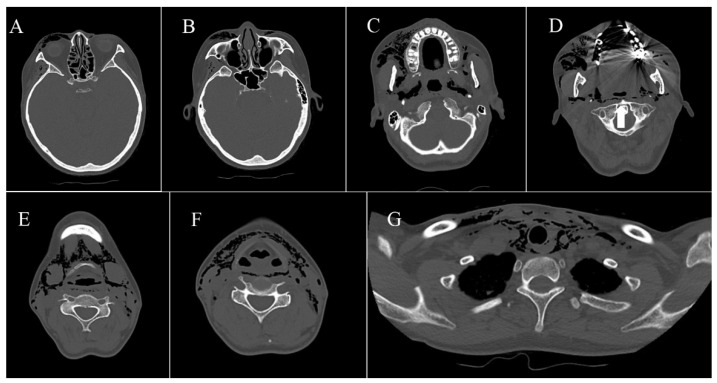
(**A**) Air was noted in the orbital soft tissue. (**B**) Air was noted in the buccal space. (**C**) Air was noted in the masticator muscle space. (**D**) Air was noted in the parotid gland space, (**E**) Air was noted in the submandibular space and the submental space. (**F**) Air was noted in the anterior cervical space and the posterior cervical space. (**G**) Air was noted in the mediastinum.

**Figure 5 healthcare-10-00290-f005:**
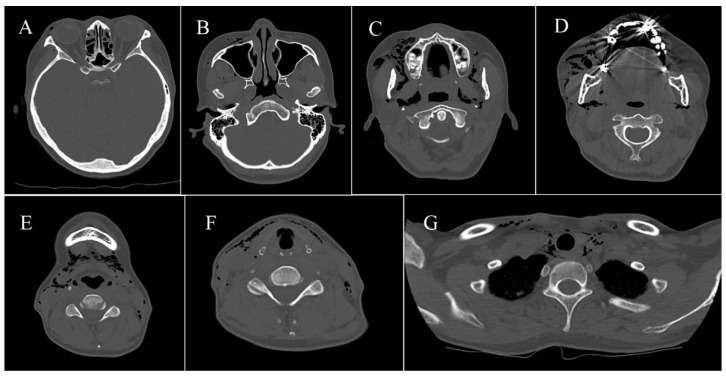
(**A**,**B**) There is little air in the orbital soft tissue and buccal space. (**C**,**D**) Air in the masticatory muscle space and the parotid gland space was slightly reduced. (**E**–**G**) Air in the submandibular space and the submental space, the mediastinum was significantly reduced.

**Figure 6 healthcare-10-00290-f006:**
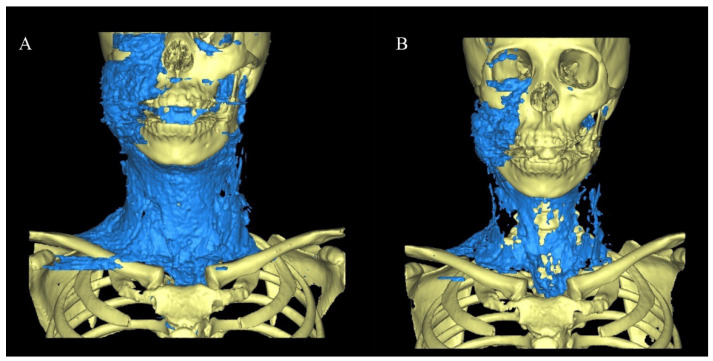
(**A**) 3D model based on the CT taken at the first visit. The volume was measured to be about 713 mL. (**B**) 3D model on the 3rd day after injury. The volume was measured to be about 248 mL.

**Figure 7 healthcare-10-00290-f007:**
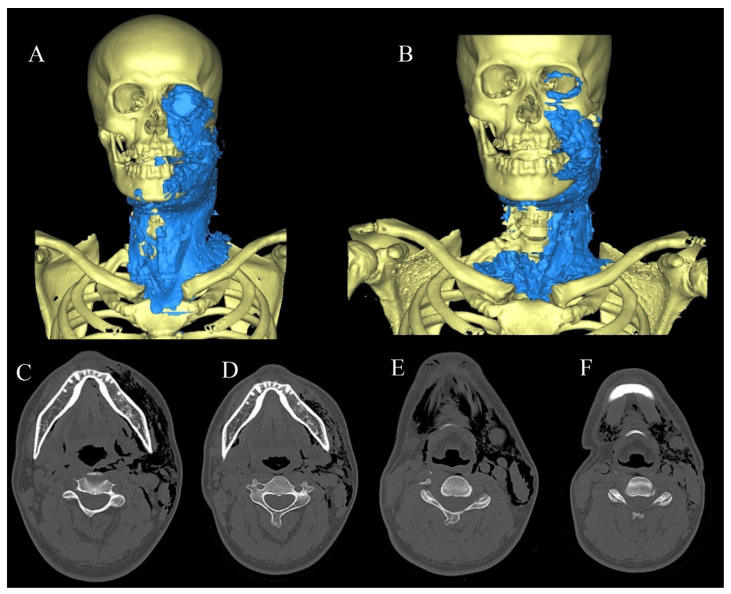
(**A**) 3D model based on the CT taken at the first visit. The volume was measured to be about 376 mL. (**B**) 3D model on the 3rd day after injury. The volume was measured to be about 204 mL. (**C**,**D**) the CT taken at the first visit. (**E**,**F**) CT findings 3 days after injury.

**Figure 8 healthcare-10-00290-f008:**
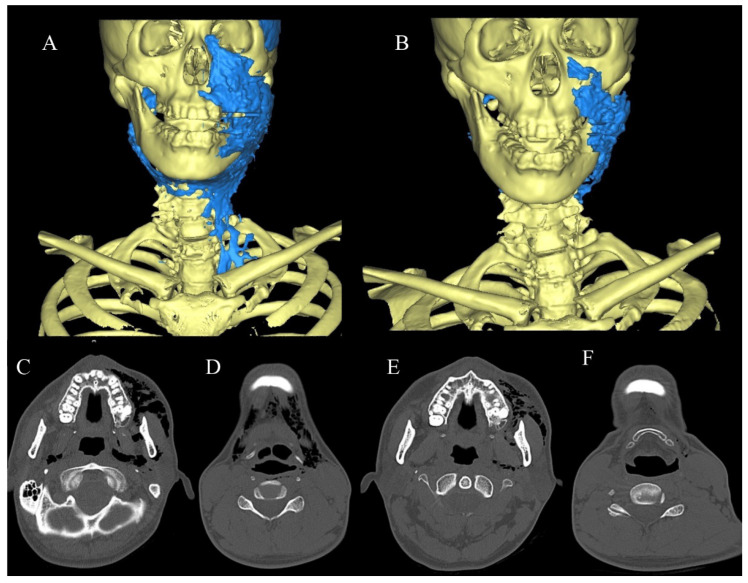
(**A**) 3D model based on CT at the first visit. The volume was measured at approximately 108 mL. (**B**) 3D model on the 3rd day after injury. The volume was approximately 24 mL. (**C**,**D**) CT at the first visit. (**E**,**F**) CT findings 2 days after injury.

**Table 1 healthcare-10-00290-t001:** CT at the first visit. The spread of emphysema to the tissue gap in cases 1–3 is shown separately at the facial, cervical, and mediastinal levels; this shows the air volume in the 3D model on the initial visit and 3rd day after the injury.

		Facial				Neck				Mediastinum				
		Orbital Soft Tissue	Buccal	Masticatory Muscle	Parotid Gland	Parapharyngeal	Posterior Pharyngeal	Sublingual	Submandibular	Anterior Cervical	Posterior Cervical	Posterior Pharyngeal	Carotid Artery Space	Anterior Mediastinum
Case1	Day 1	●	●	●	●	●	●	●	●	●	●	●	●	●
	Day 3	●	●	●		●			●	●	●		●	●
Case 2	Day 1	●	●	●	●	●	●	●	●	●		●	●	●
	Day 3	●	●	●	●	●			●	●			●	●
Case 3	Day 1		●	●	●	●	●	●	●	●				●
	Day 3		●	●	●				●					

●: Presence of subcutaneous emphysema.

## Data Availability

The data used to support the findings of this study are available from the corresponding author upon request.

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
