# Peer review of "Subcutaneous Emphysema Related to Dental Treatment: A Case Series"

_healthcare, 2022, doi:10.3390/healthcare10020290_

Round 1
Reviewer 1 Report
Dear Authors,
The case reports are interesting and well-written.
In the present version of the manuscript, the introduction is rather poor and does not contextualize your study appropriately, i.e., it does not provide enough information about etiology of cervicofacial subcutaneous Emphysema etc
- Title: I recommend that the authors indicate in the title that in contains cases, to avoid confusion and to enable the readership to find it easier.
- Abstract: Please use „The first case”, „Second case” instead of Case1, Case2…
- Introduction:
The introduction in its present form (8-10 rows of text) is insufficient to contextualize the topic of the paper, especially as the mansucript is a list of cases. You should at least double the lenght of the introduction
The Introduction should be complemented and extended with information on the epidemiology (e.g., prevalence) of SE in dental practice, both from dental treatments and from other causes (E.g., trauma) and a list of the main internventions associated with SE.
L37: please report on the advantages and disadvantages of using CT for diagnosis of SE
L37-38: was there any published evidence (EBM) on the correlation of air volume and disease severity and disease progression? if so, please include it, if not, please state that
L40: you are already stating the aims, while we still dont know why the study would be relevant. as suggested previously
L43: on a computer simulation
Case 2:
L124: is the bold necessary?
L130: Because the patient…
L134: Six days later,
The presentation of Table 1 is very poor, I recommend that the authors provide a more clear table (preferably with colors) to improve the clarity and readability of the summarized data
Discussion:
L181: it is sufficient to administer
Please discuss the overlap between prophylactic antibiotic use for dental procedures and the prevention of SE!
Summary, after these small modifications I suggest this study for publishing, because it provides useful information for dental practitioners.
Author Response
Responses to Reviewers’ Comments
Thank you very much for your invaluable comments and kind acceptance. We have incorporated all the reviewers’ comments and suggestions into our manuscript; the corresponding changes are highlighted in red font in the revised manuscript.
We would like to say thank you once again for the suggestions, which were very helpful in further improving our manuscript.
Comments from Reviewers and Responses
Reviewer 1
In the present version of the manuscript, the introduction is rather poor and does not contextualize your study appropriately, i.e., it does not provide enough information about etiology of cervicofacial subcutaneous Emphysema etc
Summary, after these small modifications I suggest this study for publishing, because it provides useful information for dental practitioners.
Response:
Thank you for your invaluable comments. As you pointed out, we have revised the points to be pointed out for the enhancement of the introduction and many improvements.
Comment 1) Reviewer1: Title: I recommend that the authors indicate in the title that in contains cases, to avoid confusion and to enable the readership to find it easier.
Response:
We thank you for this helpful comment. As you pointed out, the title has been changed to ‘Subcutaneous emphysema related to dental treatment: A Case Series.’.
Comment 2) Reviewer1: Abstract: Please use „The first case”, „Second case” instead of Case1, Case2…
Response:
We thank you for this helpful comment. We changed'case1','case2', and'case3'in the abstract to'The first case','Second case', and'Third case', respectively.
Comment 3) Reviewer1: Introduction:
The introduction in its present form (8-10 rows of text) is insufficient to contextualize the topic of the paper, especially as the mansucript is a list of cases. You should at least double the lenght of the introduction
The Introduction should be complemented and extended with information on the epidemiology (e.g., prevalence) of SE in dental practice, both from dental treatments and from other causes (E.g., trauma) and a list of the main internventions associated with SE.
Response:
In the introduction, we have added the following text citing the literature on dental treatment and other causes of subcutaneous emphysema; "Cervicofacial subcutaneous emphysema (SE) is known to be caused by the invasion of gas into the subcutaneous tissue. It’s also caused by trauma such as facial fractures, intraoral trauma, traumatic destruction of the chest wall or airway gastrointestinal tract and dental treatment. [1-4]. Jones A. conducted a literature review from 1993 to 2020 and found that dental extraction often preceded the onset of SE (54% of cases) in dental treatment. Most cases were iatrogenic, 51% due to air turbines and 9% due to air syringes, factors such as nose-blowing accounted for 10%. When gas enters the subcutaneous tissue gap, its spread rate is high, and is accompanied by swelling of the face and neck, along with dysphagia, chest tightness, and dyspnea in some cases. [5] '
Comment 4) Reviewer1: L37: please report on the advantages and disadvantages of using CT for diagnosis of SE
Response:
Thank you for this helpful comment. We have added a citation and explanation of the advantages and disadvantages of using CT.
Comment 5) Reviewer1: L37-38: was there any published evidence (EBM) on the correlation of air volume and disease severity and disease progression? if so, please include it, if not, please state that
Response:
Thank you for this helpful suggestion. Until now, there have been no published studies on the correlation between air volume and disease progression. We have added an explanation to the introduction about it.
Comment 6) Reviewer1: L40: you are already stating the aims, while we still dont know why the study would be relevant. as suggested previously
Response:
Thank you for this helpful comment. One of the purposes of this study is to evaluate the healing process of subcutaneous emphysema from the viewpoint of volume. I added the following about that point; ‘Understanding the course of the healing of SE from the evaluation item of volume over time is an important point that can serve as a guideline for future treatment.’
Comment 7) Reviewer1: L43: on a computer simulation
Response:
Thank you for this helpful comment. We added ‘on a computer simulation’.
Comment 8) Reviewer1: L124: is the bold necessary?
Response:
Thank you for this helpful comment. It's a mistake in our editing. We fixed it normally.
Comment 9) Reviewer1: “L130: Because the patient…
Response:
Thank you for this helpful comment. As you pointed out, we have fixed it.
Comment 10) Reviewer1: L134: Six days later,
Response:
Thank you for this helpful comment. Thank you for this helpful comment. As you pointed out, we have fixed it.
Comment 11) Reviewer1: The presentation of Table 1 is very poor, I recommend that the authors provide a more clear table (preferably with colors) to improve the clarity and readability of the summarized data
Response:
Thank you for this helpful comment. We have changed the layout of the table to make it easier for readers to read.
Comment 8) Reviewer1: L181: it is sufficient to administer
Response:
Thank you for this helpful comment. As you pointed out, we have fixed it.
Comment 9) Reviewer1: Please discuss the overlap between prophylactic antibiotic use for dental procedures and the prevention of SE!
Response: Thank you for your very valuable suggestions. We added the following about antimicrobial prophylaxis for SE; "Regarding resting antibiotics and infection prevention subcutaneous emphysema associated with dental procedures can cause the oral flora to invade the subcutaneous tissue due to the destruction of the oral mucosa, causing cellulitis or abscess formation during infection. Therefore, as with the administration of prophylactic antibiotics by dental treatment, prophylactic administration of antibiotics as a first choice is considered essential as a preventive measure against SE infection. [15,16,17]."
Reviewer 2 Report
The Authors raised a topic of very rare complications after dental treatment. The subcutaneous emphysema (SE) is shown to be subcutaneous tissue gas which volume can be estimated from the computed tomography (CT). The Authors tried to observe the healing process of CE caused by dental treatment using CT and to evaluate the volume changes ohf air within tissues with the use of simple tresholding. However, the major issue is the number of cases. Authors presented only three cases and in my opinion it is not possible to draw any conclusions on such a small sample. The manuscript should be classified as case series study, rather then original article. The title should contain the “case series” statement.
The manuscript contains also several minor issues:
- What CT scanner was used (type, manufacturer)?
- What were CT exposure parameters?
- From a dental point of view, it is hard for me to imagine how such large SEs could have appeared in the presented cases after dental treatment. Case 1 concerns a patient after endodontic treatment, however, no detailed description of the procedure was provided. I would like to know what canal rinsing procedure was provided, was CHX, hypochlorite or hydrogen peroxide used and in what concentration were used. Was laser induced irrigation used?
- Case 2: How the procedure of extraction was performed? Was PRF used? Was the extraction accompanied by the removal of the cyst or other inflammatory lesion? What type of anesthesia was used? Was excessive bleeding or suturing used?
- Case 3: Was dental air polishing sandblaster used? Did the patient have periodontal problems?
- Figures 5, 6AB, 7AB are unclear. The segmentation results should be presented on horizontal CT slices with outlined borders of segmented air and tissues. On the basis of the presented VR reconstruction, I have a suspicion that apart from the air inside the tissues, the segmentation also includes air outside the body or in the airways.
- The formatting of the manuscript should be corrected.
- The topic should be changed from “Virtual Reality, Digital Twins, the Metaverse” to more adequate.
- Extensive editing of English language and style are required.
Author Response
Responses to Reviewers’ Comments
Thank you very much for your invaluable comments and kind acceptance. We have incorporated all the reviewers’ comments and suggestions into our manuscript; the corresponding changes are highlighted in red font in the revised manuscript.
We would like to say thank you once again for the suggestions, which were very helpful in further improving our manuscript.
Comments from Reviewers and Responses
Reviewer 2
The Authors raised a topic of very rare complications after dental treatment. The subcutaneous emphysema (SE) is shown to be subcutaneous tissue gas which volume can be estimated from the computed tomography (CT). The Authors tried to observe the healing process of CE caused by dental treatment using CT and to evaluate the volume changes ohf air within tissues with the use of simple tresholding. However, the major issue is the number of cases. Authors presented only three cases and in my opinion it is not possible to draw any conclusions on such a small sample. The manuscript should be classified as case series study, rather then original article. The title should contain the “case series” statement.
Response:
We thank you for this helpful comment. As you pointed out, the title has been changed to ‘Subcutaneous emphysema related to dental treatment: A Case Series.’.
Comment 1) Reviewer2: What CT scanner was used (type, manufacturer)?
Response:
We have added details such as CT scanner products as follows; Using a non-enhanced multidetector CT scanner (Aquilion; Toshiba Medical Systems, Tokyo, Japan)
Comment 2) Reviewer2: What were CT exposure parameters?
Response:
Thank you for your valuable comments.
The CT of this study was taken with Tube voltage 120kVp, tube current 100mA.
We added it in the text.
Comment 3) Reviewer2: Case 1 concerns a patient after endodontic treatment, however, no detailed description of the procedure was provided. I would like to know what canal rinsing procedure was provided, was CHX, hypochlorite or hydrogen peroxide used and in what concentration were used. Was laser induced irrigation used?
Response:
Thank you for your valuable comments. Root canal treatment was also combined with alternate irrigation with hydrogen peroxide. We added it in the text.
Comment 4) Reviewer2: Case 2: How the procedure of extraction was performed? Was PRF used? Was the extraction accompanied by the removal of the cyst or other inflammatory lesion? What type of anesthesia was used? Was excessive bleeding or suturing used?
Response:
Thank you for your valuable comments. An air turbine was used during tooth extraction. We added it in the text.
Comment 5) Reviewer2: Case 3: Was dental air polishing sandblaster used? Did the patient have periodontal problems?
Response:
Thank you for your valuable comments. Periodontal disease was also present in his oral environment. No sandblasting was used to treat him.
Comment 6) Reviewer2: Figures 5, 6AB, 7AB are unclear. The segmentation results should be presented on horizontal CT slices with outlined borders of segmented air and tissues. On the basis of the presented VR reconstruction, I have a suspicion that apart from the air inside the tissues, the segmentation also includes air outside the body or in the airways.
Response:
Thank you for your very important comment. We will show you how to select only subcutaneous emphysema in this study. If a site different from subcutaneous emphysema such as the airway was selected, it was manually removed. A method for measuring subcutaneous emphysema using 3D simulation from CT data has been added to the supplementary materials. Please check.
Reviewer 3 Report
1.Well articulated manuscript,however,the quality of written english can be further improved.
2.There are a few long sentences throughout the manuscript which can be broken down to 2 or more smaller sentences for a better understanding.
3.It would have been desirable if a few clinical photographs of the patient was added into the manuscript.
4.What was the experience of the radiologist who had interpreted the CT scan findings?
5.A note on the complications in patients who were untreated can be added.
6.A note on other investigative and treatment modalities for cases of emphysema along with their advantages & disadvantages can be added.
Author Response
Responses to Reviewers’ Comments
Thank you very much for your invaluable comments and kind acceptance. We have incorporated all the reviewers’ comments and suggestions into our manuscript; the corresponding changes are highlighted in red font in the revised manuscript.
We would like to say thank you once again for the suggestions, which were very helpful in further improving our manuscript.
Comments from Reviewers and Responses
Reviewer 3
The Authors raised a topic of very rare complications after dental treatment. The subcutaneous emphysema (SE) is shown to be subcutaneous tissue gas which volume can be estimated from the computed tomography (CT). The Authors tried to observe the healing process of CE caused by dental treatment using CT and to evaluate the volume changes ohf air within tissues with the use of simple tresholding. However, the major issue is the number of cases. Authors presented only three cases and in my opinion it is not possible to draw any conclusions on such a small sample. The manuscript should be classified as case series study, rather then original article. The title should contain the “case series” statement.
Response:
We thank you for this helpful comment. As you pointed out, the title has been changed to ‘Subcutaneous emphysema related to dental treatment: A Case Series.’.
Comment 1) Reviewer3:. Well articulated manuscript,however,the quality of written english can be further improved.
2.There are a few long sentences throughout the manuscript which can be broken down to 2 or more smaller sentences for a better understanding.
Response:
Thank you for your valuable comments.
We reviewed the entire manuscript and were again checked for native English proofreading.
Comment 2) Reviewer3: It would have been desirable if a few clinical photographs of the patient was added into the manuscript.
Response:
Thank you for your valuable comments.
We have added clinical photos for Case 1.
Comment 3) Reviewer3: What was the experience of the radiologist who had interpreted the CT scan findings?
Response:
Thank you for your valuable comments. CT interpretation was judged by oral and maxillofacial surgeons at the 7th and 15th years of the head and neck. The chest was determined by a veteran radiologist with a specialist.
Comment 4) Reviewer3: A note on the complications in patients who were untreated can be added.
Response:
Thank you for your valuable comments. We have added a review of the literature regarding the complications of patients who have not been treated with SE; Resting the patient through hospitalization management is desirable. If hospitalization is difficult, careful observation of the airways and non -monitoring of gas progression may lead to widespread emphysema and dyspnea due to airway compressions. In addition, if it spreads to the mediastinum, it may affect respiratory function and cardiac function, so caution is required. [14]
Comment 5) Reviewer3: A note on other investigative and treatment modalities for cases of emphysema along with their advantages & disadvantages can be added.
Response:
Thank you for your valuable comments. Periodontal disease was also present in his oral environment. No sandblasting was used to treat him.
Round 2
Reviewer 2 Report
Authors have addressed most of my concerns. The article would bring much more and have higher impact if the research were carried out on a larger group of patients. Furthermore, the quality of air segmentation is beyond assessment. Besides, I have no more comments.